# Rheological Characterization and Quality of Emulsions Based on Fats Produced during the Reaction Catalyzed by Immobilized Lipase from Rhizomucor Miehei

**Małgorzata Kowalska** [1,*] , **Marcin Krzton-Maziopa** [2,3], **Anna Krzton-Maziopa** [4] , **Anna Zbikowska** [5] **and Jerzy Szakiel** [6]

1 Faculty of Chemical Engineering and Commodity Sciences, Kazimierz Pulaski University of Technology and Humanities, Chrobrego 27, 26-600 Radom, Poland
2 Faculty of Electronics and Information Technology, Warsaw University of Technology, Nowowiejska 15/19, 00-665 Warsaw, Poland; marcin.krzton-maziopa.stud@pw.edu
3 Faculty of Production Engineering, Warsaw University of Technology, Narbutta 85, 02-524 Warsaw, Poland
4 Faculty of Chemistry, Warsaw University of Technology, Noakowskiego 3, 00-664 Warsaw, Poland; anka@ch.pw.edu.pl
5 Department of Food Technology and Assessment, Institute of Food Sciences, Warsaw University of Life Sciences-SGGW (WULS-SGGW), 02-772 Warsaw, Poland; anna_zbikowska@sggw.edu.pl
6 Department of Non-Food Product Quality and Safety, Cracow University of Economics, Rakowicka St. 27, 31-510 Cracow, Poland; szakielj@uek.krakow.pl
* Correspondence: mkowalska7@vp.pl

**Abstract:** It has been shown that structured lipids, formed in the process of enzymatic modification of natural hard fat with walnut oil, are capable of stabilizing emulsion systems without the need to add additional emulsifiers. This is especially true for emulsions containing fat formed during enzymatic modification when the amount of added water to the reaction catalyst was in the range of 12–16 wt%. Physicochemical evaluations, i.e., the average particle size, its growth, distribution, and dispersity coefficient, were comparable with the reference emulsion where the emulsifier was lecithin, well-known for its emulsifying properties. Microstructure studies also confirmed the above observations. Rheological studies performed on a set of emulsions containing structured lipids of variable composition confirmed that interesterified lipid blends can be directly utilized as a fat base in the preparation of stable emulsions. The consistency, thixotropic behavior, long-term shelf life, and thermal stability of these emulsions were found to be comparable to systems stabilized with conventional emulsifiers, i.e., sunflower lecithine. Our approach offers the opportunity for the preparation of stable emulsion systems, free from additional emulsifiers, for the food or cosmetics industry, which is extremely important from the point of view of the preparation of products free from allergens.

**Keywords:** rheological structure of the emulsion; quality of the emulsion; structured lipids; enzymatic modification

## 1. Introduction

Growing interest in new types of bio-compatible emulsions based on natural raw materials for food and cosmetics applications stimulated intensive scientific efforts on the intensively developing field of structured lipids. This broad group of materials includes: triacylglycerols modified by the incorporation of new fatty acids into their structure, reorganized to change the positions of fatty acids or the fatty acid profile regarding the natural state, or novel lipids not occurring naturally. The main advantage of the structural or compositional modification of lipids is closely related to the possible health benefits resulting, e.g., from the "diet-friendly" lipids for the whole population or better compatibility with the nutritional restrictions of patients with exocrine pancreatic insufficiency

or various problems associated with the impaired assimilation of fats. Comprehensive investigations on structured lipids carried out at the end of the 1980s showed that modified lipids containing a caprylic group at the 1- and 3-positions and a long-chain fatty acid-based functionality located in the 2-position in the glycerol molecule underwent faster hydrolysis and were assimilated more efficiently than conventional lipids based on long-chain triacylglycerols [1]. Modified lipids, especially those containing diverse fatty acids incorporated into the glycerol backbone, open new possibilities in the field of cosmetics or foodstuff formulations, where tailor-made structured lipid-based emulsions are essential to achieve desirable skin nutrition and humidification or are dedicated to a part of the population suffering a specific problem with fats assimilation. Moreover, structured lipids due to their emulsifying abilities can be directly applied in the production of emulsions also as emulsifiers, allowing the suitable consistency and stability of emulsions, which eliminates the necessity of the utilization of other emulsifying agents [2].

Rheological properties of emulsions, such as viscosity, smoothness, and granularity, determine the perception of touch, taste, or flavor. These properties influence significantly the long-term stability of emulsion systems and are essential during their processing, transportation, and everyday application [3]. Studies of the rheological properties of food emulsions containing structured lipids based on caprylic acid and olive oil have shown considerable improvements in the final consistency of products and evidenced their long-term stability in comparison to standard formulations [4]. In addition, emulsions containing modified lipids were found to be more stable and exhibited better resistance to thermal shock caused by rapid warming after removal from cooled storage to ambient temperature. Investigations on the relationship between the structure of the modified lipids and their rheological properties revealed that the viscosity of emulsions based on these lipids change proportionally to the length of the fatty acid chains present in their molecules [5]: the longer the chains in the modified triacylglycerol (TAG), the higher the viscosity of the emulsion. This opens the way for further modifications of natural lipids and for the synthesis of materials with dedicated properties for specific application. As the emulsion consistency and its long-term stability are the most important parameters determining industrial application, the main goal of this work was showing the method of synthesis of low-cost multifunctional materials, allowing for the preparation of stable food, cosmetics, or pharmaceutical formulations. Apparently, the use of the structured lipids could be beneficial here as those materials can be relatively easily modified and utilized directly as new emulsifiers. In a typical emulsion system, the kinetic stability of the system is achieved by the appropriate choice of the type and adjustment of the quantity of an emulsifier added during the homogenization process [6–8]. As the effective emulsifier is required to quickly adsorb on the surface of the oil droplets and concomitantly reduce the interfacial tension by the formation of a thin film preventing the aggregation of neighboring oil droplets, the amount of emulsifier used for preparation should be sufficient to cover the surfaces of all the oil droplets [9,10]. Essentially, the size of droplets formed during homogenization depends on two processes involving the initial generation of droplets of small size and their rapid stabilization against coalescence, once they are formed [11]. From a technological point of view, the replacement of standard emulsifying agents by a multifunctional structured lipid would enhance both the stability of the emulsion and its performance. Rheological properties of the emulsion system are often treated as an indicator of the product quality just after preparation and after a few weeks of storage [12–17]. Dynamic and oscillatory rheological tests performed under controlled conditions provide crucial information about the emulsions stability and their response to various stimuli (temperature, shearing, oscillatory strain, pH, etc.) and allow fast evaluation of the stability of a particular emulsion system [18–22].

Interesterification involves a series of reactions involving the exchange of acyl groups [23] between an ester and an alcohol (alcoholysis), an ester and an acid (acidolysis), and an ester

and an ester (interesterification, transesterification, esterification proper) according to the following equation (Equation (1)).

$$
\begin{bmatrix} A \\ A \\ A \end{bmatrix} + \begin{bmatrix} B \\ B \\ B \end{bmatrix} \rightleftarrows \begin{bmatrix} A \\ A \\ B \end{bmatrix} + \begin{bmatrix} B \\ A \\ B \end{bmatrix} + \begin{bmatrix} B \\ B \\ A \end{bmatrix} + \dots \tag{1}
$$

Technologically, the exchange reaction between esters, mainly between triacylglycerols, is the most important [24]. This exchange can take place between triacylglycerol molecules, which is then intermolecular interesterification, or if fatty acid residues are exchanged inside the triacylglycerol molecule, which is then intramolecular interesterification. One way of modifying fats is the interesterification process using enzymes as reaction biocatalysts. According to [25], enzymatic interesterification has received considerable attention in recent years to replace chemical interesterification. Increased interest for that method is related to the design of other innovative fats. Enzymes used in this reaction catalyze hydrolysis reactions of triacylglycerols and incomplete acylglycerols (MAG, DAG). The direction of this reaction was used in the presented work to produce a fat containing in its composition an optimized amount of the mentioned emulsifiers. Hydrolysis is a reversible reaction (according to Equation (2)); therefore, under conditions of a limited amount of water in the reaction system, the synthesis of acylglycerols will take place in the opposite direction

$$
\begin{bmatrix} A \\ A \\ A \end{bmatrix} + H_2O \rightleftarrows \begin{bmatrix} OH \\ A \\ A \end{bmatrix} + \begin{bmatrix} OH \\ A \\ OH \end{bmatrix} + AH \tag{2}
$$

The aim of this study was to investigate the set of emulsions containing structurally modified oil phases composed of interesterified fat blends with enhanced amounts of monoacylglycerols (MAGs) and diacylglycerols (DAGs), obtained through the enzymatic interesterification [23] of mutton tallow and walnut oil in the weight ratio (1:1). Enzymatic interesterification facilitates the incorporation of unsaturated fatty acids of walnut oil into the triacylglycerol (TAG) structures of mutton tallow (MT). Mutton tallow is a hard animal fat, which contains in its structure conjugated linoleic acid (CLA), and the combination of MT with walnut oil would result in products containing CLA and an amplified amount of unsaturated fatty acids. These new structured lipids containing both types of fatty acids offer a unique combination for possible application in, e.g., the cosmetics and food industry, where they can be utilized as an ingredient of novel emulsions free from synthetic emulsifiers. To reveal the stability and consistency of the cosmetic emulsions containing novel structured lipids, a comprehensive study of their rheological properties, covering the dynamic and oscillatory tests, was carried out. To illustrate the benefits coming from the application of structured lipids as effective emulsifying agents, in this paper, the rheological properties of emulsions based on interesterified lipids were compared to the corresponding parameters of conventional emulsions stabilized with sunflower lecithin.

## 2. Materials and Methods

### 2.1. Materials

Raw mutton tallow ("Meat-Farm" R. Łuczak, Wolka Kosowska, Poland) was bleached and deodorized in a laboratory under vacuum at elevated temperature (105 °C) prior to use. The extra virgin walnut oil (AmanPrana, Perigord, France) was used without further purification. Lipozyme RM IM (Novozymes, Bagsvaerd, Denmark) with an activity of 5–6 Baun/g (Baun—Batch Acidolysis Novo Unit Activity determines the enzyme activity relative to 1 g of the enzymatic preparation) was used as a biocatalyst for interesterification. The enzymatic preparation consists of immobilized lipase from Rhizomucor miehei and, for activation, it requires the addition of at least 4 wt.% of water before use in the reaction. Sunflower lecithin (Lasenor Emul, S.L. Barcelona, Spain) served as an emulsifier in the reference emulsion. All the prepared emulsions contained hydroxypropylmethylcellu-

lose (Barentz Company, Hoofddorp, The Netherlands) as a thickening agent and sodium benzoate (Orff Food Eastern Europe) as a preservative.

*2.2. Methods*

2.2.1. Interesterification of Mutton Tallow with Walnut Oil

Before interesterification, a mixture composed of bleached and deodorized mutton tallow and walnut oil in 1:1 weight proportions was homogenized at 70 °C in an inert atmosphere to prevent the oxidation of reagents. Afterward, the 50 g samples of the freshly prepared fat mixture were transferred to four separate flasks, placed in the plate-shaker thermostat, warmed to 60 °C, and equilibrated over one hour. Next, 8 wt.% of biocatalyst (immobilized lipase) was added to each flask. Bearing in mind that the quantity of monoacylglycerols (MAGs) and diacylglycerols (DAGs) in the final product is closely related to the hydrolysis process, the following amounts of water: 10, 12, 14, 16 wt%, were added to the enzymatic preparation directly before reaction. After adding water, the interesterification process was carried out for 6 h and finished by filtering off the enzymatic preparation. After synthesis, the obtained interesterified oil phases were transferred into sealed containers, carefully characterized with regard to their chemical composition, fatty acid profiles, and polar fraction content, and in a further step, utilized for the preparation of emulsions.

2.2.2. Determination of Polar Fraction Content

The amount of polar fraction in the oil phase plays a crucial role in the formation of a stable emulsion system. Our aim was to obtain the oil phases with enhanced amounts of polar fractions; therefore, in all the prepared materials, the amounts of polar phases were determined just after interesterification. To this end, the prepared fat blends were separated into nonpolar and polar fractions using the column chromatography technique (column filled with silica gel SG 60, 70–230 mesh, Merck, Germany) and then characterized further to obtain information about the structure of the prepared materials. The nonpolar fraction, containing mainly triacylglycerols (TAGs), was eluted with a solvent mixture containing petroleum ether (87 vol.%) and diethyl ether (13 vol.%). For the separation of polar fraction, composed of free fatty acids (FFA), monoacylglycerols (MAGs), and diacylglycerols (DAGs), ethyl ether was used as an eluent. The contents of TAGs and polar fractions in the prepared oil phases were determined after evaporation of the eluent according to the ISO 8420 standard [26].

2.2.3. Preparation of Emulsions

Emulsions containing the structured lipids as oil phases were prepared according to the method published elsewhere [27] using the following amounts of ingredients: oil phase—30 wt.%, hydroxypropylmethylcellulose—1 wt.%, sodium benzoate—0.25 wt.%. Distilled water was used to adjust the quantity of all ingredients to 100 wt.%. Emulsions based on interesterified lipids were prepared without additional emulsifiers. A reference emulsion with a noninteresterified fat blend and 5 wt.% of sunflower lecithin as an emulsifier was also prepared for comparison. To prepare emulsions for further studies, both aqueous and oil phases were warmed to 55 °C, mixed together, and homogenized using homogenizer SILVERSON L4R for 3.5 min at 18,000 rpm (rotations per minute). Afterward, the materials were cooled down to room temperature, sealed in tight containers, and stored in a refrigerator (8–10 °C).

2.2.4. Droplet Size Distribution

The concentration of droplets in an emulsion is usually described in terms of the dispersed-phase volume fraction, which is equal to the volume of emulsion droplets divided by the total volume of the emulsion. The knowledge of the dispersed-phase volume fraction and droplet size distribution are important from the technological point of view as both affect the appearance, texture, flavor, stability, and cost of emulsion products. Many of the

most important properties of emulsion products (e.g., shelf life, appearance, texture, or flavor) are determined by the size of the droplets they contain [28,29]. Consequently, it is important for technologists and scientists to be able to reliably control, predict, measure, and report the size of the droplets in emulsions.

The average particle size and their distribution were determined after 72 h and 2 months after preparation at 22 °C. For each measurement, the emulsions were diluted with distilled water in the ratio of 1:200. Droplet size was measured in the range of 0.12–704 μm by laser scattering using a Microtrac Particle Size Analyzer (Leeds & Northrup, Roanoke, Virginia, USA). The time of determination of each sample was about 0.5 min. Determinations were performed in three parallel replications for each sample and calculated as an average value. The dispersion index (K) for each emulsion was calculated from Equation (3):

$$K = \frac{D_a - D_b}{D_c} \tag{3}$$

where $D_a$, $D_b$, and $D_c$ are droplet diameters (μm) taken from the differential volumes of 90, 50, and 10% of particles that have diameters lower than the stated value, respectively.

### 2.2.5. Microstructure Evaluation of Emulsion Systems

The microstructure of the emulsion systems 72 h after fabrication was analyzed using a Genetic Pro Trino optical microscope (Delta Optical, Warsaw, Poland) and a DLT Cam Pro camera (Delta Optical, Warsaw, Poland). The images presented in this paper were taken using a total magnification of G × 400.

### 2.2.6. Rheological Tests

Viscoelastic properties of emulsions based on structured lipids were performed at the Faculty of Chemistry, Warsaw University of Technology using a Physica MCR 301 rheometer equipped with parallel plate-measuring geometry (plate diameter of 50 mm and 1 mm measuring gap) and a built-in Peltier device enabling precise temperature control. Dynamic tests in controlled shear stress (CSS) mode were performed to reveal the rheological behavior of the studied materials under the shear flow conditions. To study the changes in rheological parameters before and after the prolonged storage period, small-amplitude oscillatory experiments were carried out in the linear viscoelastic region (LVR) of as-prepared and stored emulsions. For the assessment of emulsions stability after storage, the changes in mechanical moduli (G'-storage and G''-loss modulus) with frequency were performed in an angular frequency range varying from 0.1 to 100 rad/s at a constant strain amplitude (0.5%). All rheological studies were carried out at a test temperature of 20 °C.

### 2.2.7. Statistical Analysis

The results were subjected to one-way ANOVA. The Tukey test was employed to determine statistically significant differences. The level of $p < 0.05$ was considered significant. Statgraphics plus 4.0 package (Statistical Graphics Corp., Warrenton, VA, USA) was used.

## 3. Results and Discussion

During the enzymatic interesterification of lipids, two reactions take place simultaneously: hydrolysis and rearrangement of triacylglycerols (TAGs), which results in a mixture of products composed of TAGs, monoacylglycerols (MAGs), diacylglycerols (DAGs), and free fatty acids (FFAs) [30]. The competitive hydrolysis reaction reduces the final yield of interesterified TAGs and significantly accelerates the interesterification reactions, leading finally to a higher concentration of MAGs and DAGs, which are essential intermediates [31]. In our syntheses, by the addition of suitable amounts of water to the enzymatic catalyst, we intentionally shifted the position of the reaction equilibrium to induce the formation of a higher amount of MAGs and DAGs during hydrolysis. This allowed us to control the amounts of the polar fraction formed in the process. The maximum amount of polar fraction (30 wt.%) was formed in a fat mixture catalyzed by Lipozyme RM IM after the

addition of 16 wt.% of water. Lower amounts of water in the reaction mixtures resulted in a decreased content of polar fraction in the final products. The contents of polar fractions in structured lipids formed via interesterification after the addition of different quantities of water are shown in Table 1.

**Table 1.** Amount of polar fractions (PFs) in the oil phases, average droplet size (ADS), and dispersion index (K) of emulsions prepared from modified lipids, determined after 72 h and after 8 weeks of storage.

| Oil Phase | Amount of Water Added to Catalyst (t.%) | PF Content (wt.%) | Emulsion Name | ADS (mv) of Fresh/Stored Emulsions (µm) | K of Fresh/Stored Emulsions |
|---|---|---|---|---|---|
| Blend 1 (physical mixture) | 0 | 0 a ± 0.00 | EM1 | 4.2 a ± 0.01/ 6.9 a ± 0.01 | 1.8/1.8 |
| Blend 2 | 10 | 18 b ± 1.65 | EM2 | 30.1 d ± 2.25/ 39.4 d ± 3.12 | 4.4 /6.9 |
| Blend 3 | 12 | 22 c ± 2.12 | EM3 | 5.1 c ± 0.01/ 15.6 c ± 1.56 | 1.8/2.3 |
| Blend 4 | 14 | 26 d ± 2.10 | EM4 | 4.7 b ± 0.02/ 14.9 b ± 1.12 | 1.8/2.3 |
| Blend 5 | 16 | 30 e ± 2.50 | EM5 | 4.9 bc ± 0.03/ 15.5 c ± 0.07 | 1.7/2.2 |

a, b, c, d: different letters indicate mean values that differ statistically significantly ($p < 0.05$).

Oil phases with different amounts of polar fractions were used for the preparation of emulsions. The quality characteristics of emulsion products are the result of complex interactions and include molecular, microscopic, macroscopic, colloidal, and organoleptic structures and characteristics [32]. Therefore, it is considered difficult to produce an emulsion product that exhibits high stability, is safe, and meets consumer requirements [33]. Depending on the application of the product, the expectations about its final quality characteristics may be different; however, the most important quality determinants of emulsion products include stability, rheological properties, and appearance [34]. Taking the above into account, such features were taken into account in the evaluation of emulsions produced on the basis of interesterified fats.

The emulsion (EM1) containing the physical mixture of lipids (Blend 1), stabilized with lecithin, was also prepared as a reference, and tested under the same conditions. Analysis of the average droplet size and their distributions, performed on studied emulsions, revealed that after the first 72 h from preparation, the emulsions based on interesterified lipids, except emulsion EM2 (based on Blend 2 containing 18 wt.% of polar fraction), were characterized by droplet sizes and distributions comparable to emulsion EM1 (stabilized with lecithin). This confirms undoubtedly their comparable stability in a short time period after preparation. In emulsion EM2, a nearly 6 times bigger average droplet size was observed with a twice higher dispersion index, which clearly indicated an insufficient amount of emulsifiers formed during the interesterification process in the oil phase. Moreover, the instability of this emulsion was also confirmed by the initial droplet size distribution showing more than two fractions (see Figure 1A,B).

After eight weeks of storage, an increase in average droplet size of all the emulsions was observed. Generally, the average particle size is influenced by the type of homogenization, homogenization time, but also by storage time and conditions (humidity and temperature). Particularly unstable temperatures can result in changes in the emulsion, including changes in the particle size of the emulsion. However, the determination of the droplet size of emulsion is the most important way of emulsion characterization, as it influ-

ences the properties of emulsion such as rheology, texture, shelf life stability, appearance, and taste [11,13].

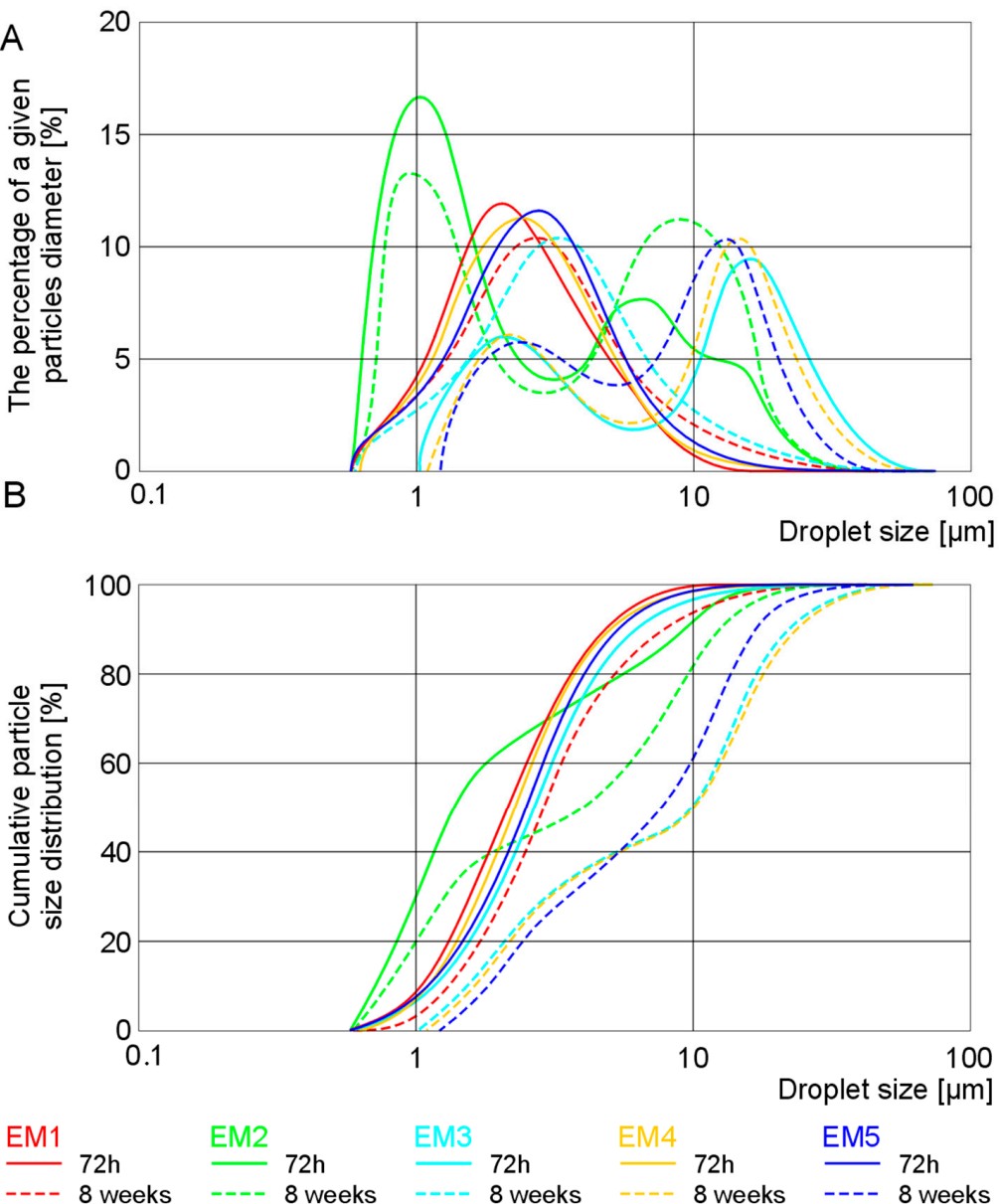

**Figure 1.** Particle size distributions of the prepared emulsions determined after 72 h and 8 weeks from preparation. (**A**) Differential curve; (**B**) cumulative curve.

The shift in differential curves in Figure 1A to the right indicates the formation of droplets with a higher average droplet size. As expected, emulsion EM2 showed the highest instability evidenced by a further enhancement of the average droplet size and a higher fraction of bigger droplets. For emulsions EM3, EM4, and EM5, a slight increase in dispersion index values, accompanied with the appearance of an additional fraction of bigger droplets, was observed. In these emulsions, about 70% of droplets had an average droplet size below 3 µm (Figure 1B), similarly to emulsion EM1 stabilized by lecithin. For EM2, the initial average droplet size was largest, reaching about 30 µm, and increased to 39 µm on storage. At the same time, the average size of droplets of three emulsions (EM3, EM4, and EM5) increased nearly three times (Figure 1A,B), while in emulsion stabilized with lecithin (EM1), the observed changes were twice smaller as the average droplet size

was found to increase about 1.6 times, with 70% of droplets having an average diameter lower than 3.7 μm.

As indicated by the author [32], larger droplets of the dispersed phase undergo the process of gravitational separation faster than smaller ones. Therefore, the probability of their mutual collision increases, which may lead to coalescence and an even greater acceleration of the system creaming. Similar observations were noticed by the authors in [35] who claimed that stable emulsion systems have smaller droplets, whereas, in accordance with Stokes' law, the speed of the phase separation process increases with their size.

Authors in [36] indicated that hydroxypropylmethylcellulose is an ineffective viscosity modifier due to the fact that it forms a network that poorly stabilizes the droplets of the dispersed phase of emulsion. On the other hand, the obtained results of stability evaluation of the systems produced on the basis of modified fats prove that the effectiveness of this thickener depends on both the composition of the fatty phase and the type of emulsifier used.

In general, emulsions EM3, EM4, and EM5 stabilized by emulsifiers produced during enzymatic modification on the basis of data on mean particle size, particle distribution, and dispersity coefficient showed comparable dynamics and stability. However, with respect to the EM1 emulsion designated as a reference emulsion in this work, the dynamics of changes in these emulsions was faster. This was evidenced by the appearance of an additional fraction after the storage time. The results recorded for the second emulsion (EM2) confirmed that, already in the first measurement, its distribution indicated that the emulsion did not show a monodisperse character, which can be translated into a lower efficiency of emulsifiers present in the system.

The measurement of the emulsion microstructure (Figure 2) also confirmed that the most homogeneous arrangement and the smallest particle size were recorded for emulsion EM1. A similar picture was recorded for emulsions EM3, EM4, and EM5. The droplets of the dispersed phase were uniformly dispersed in a continuous phase and characterized by a relatively small diameter and high symmetry. As pointed out by [37], systems with such an incoherent structure and such a chaotic arrangement of dispersed-phase droplets will exhibit trends of destabilizing changes during storage. This confirms what was indicated earlier that, in these emulsions, the effectiveness of mono and diacylglycerols as emulsifiers of emulsions was comparable to that of lecithin. For EM2 emulsions, the structure clearly showed clusters of larger droplets and much larger areas of free space.

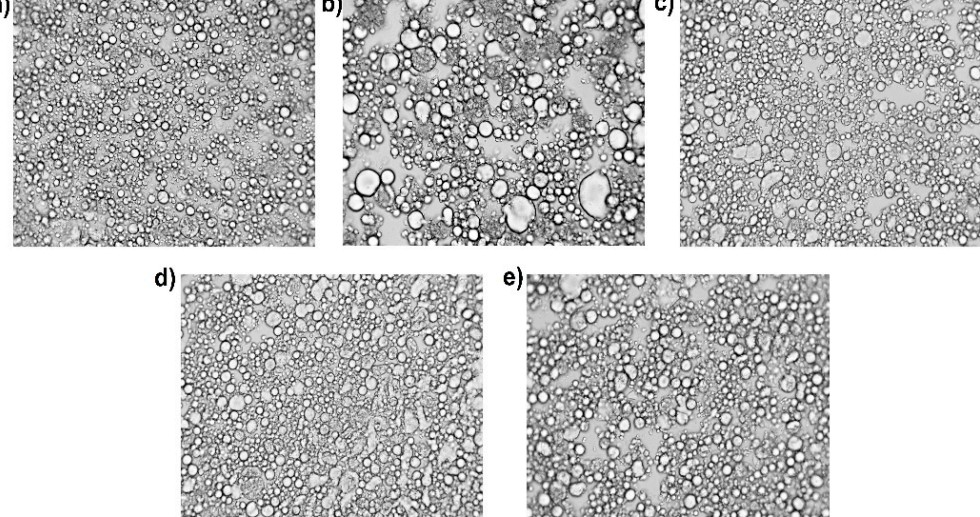

**Figure 2.** Emulsion microstructure for: EM1 (**a**), EM2 (**b**), EM 3 (**c**), EM4 (**d**), and EM5 (**e**) after 72 h from preparation (G × 400).

The observed changes in droplets morphology obviously had a great effect on the rheological behavior of the investigated emulsions manifested by the alteration in their

viscoelastic properties, which were studied via analysis of the frequency dependencies of the corresponding mechanical moduli (G'—storage modulus and G''—loss modulus) under imposed small-amplitude oscillatory strain. Oscillatory tests evidenced clearly the relation between the consistency of the investigated emulsions, droplet morphology, and the content of polar fraction in interesterified lipids used as oil-phases. As can be seen in Figure 3, all the investigated emulsions exhibited moderate frequency dependences of G' and G'' values that can be associated with the emulsion microstructure reconstruction under superimposed strain. It can also be seen that, at the same time, in all tested samples, the G' values progressively increased with the amount of polar fraction content and then leveled off, enabling studies of short-range dynamics. All the materials resembled physically cross-linked viscoelastic gels within the tested frequency range, which can be attributed to the enhanced stabilization of emulsions systems containing higher amounts of polar fractions in structured lipids. Polar fractions play the role of emulsion stabilizers and affect significantly the colloidal interactions (van der Waals forces, electrostatic, hydrophobic, steric, and hydrodynamic) between neighboring droplets, and their ability to modify the oil–water interface enables the formation of a stable emulsion [11]. Nonionic amphiphiles, such as acylglycerols (especially monoacylglycerols), exhibit high surface activity and, due to that, are widely used as emulsifiers in the preparation of various foodstuffs, cosmetics, and chemical products. The stabilization of droplets emulsion systems is held by electrostatic and steric interactions. Steric interactions are attractive in nature as they result directly from the interpenetrating and/or compression of emulsifying agents layers that cover droplet surfaces.

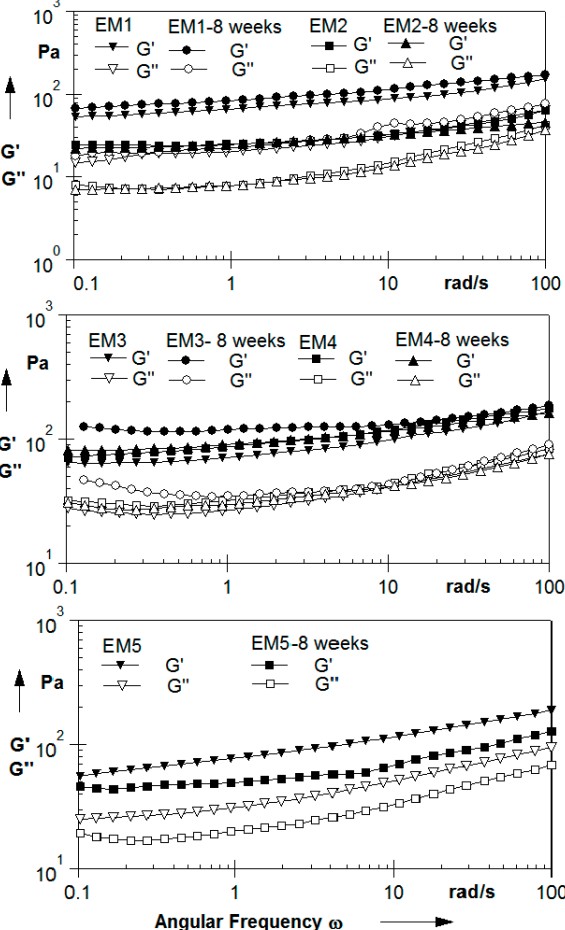

**Figure 3.** Frequency sweeps recorded for the as-prepared and stored emulsions with different amounts of polar fraction in interesterified fat (the content of polar fraction has been given in Table 1). Error bars for the measured values are within the size of point markers.

On the other hand, electrostatic ones arise from the overlapping of the electrical double layers and mostly contribute to repulsion forces between charged emulsion droplets. There are also systems that may play the dual role of electrosteric stabilizers—those systems usually comprise longer chains partially miscible with the oil phase. When using small nonionic surfactants such as polysorbate 80, one can expect that the steric contribution would be dominant while, for small ionic surfactants such as sodium lauryl sulfate or CITREM (citric acid esters of mono- and diglycerides), the electrostatic interaction will be the most pronounced. The situation becomes more complicated in the case of polymeric surfactants, such as gelatin phospholipids or pectins, as the conformation of polymer chains and formation of hydrogen bridges between neighboring droplets, carbonyl–carbonyl interactions, and dipolar forces must also be considered [38].

Stored emulsions in general were characterized by higher G′ values that can be associated with a wider droplet size distribution favoring the formation of a more rigid microstructure composed of mutually interacting droplets of different sizes. Smaller droplets tend to fill the space between bigger ones, enhancing the strength of the formed microstructure. Interestingly, the G′ of emulsion EM2 (based on Blend 2 containing 18% of polar fraction) was found to be slightly frequency-dependent. The data obtained for emulsion EM5, prepared with the interesterified lipid containing a 30% polar fraction, indicate that the higher amount of polar fraction preserves the low polydispersity index but the emulsion undergoes destabilization in the longer period of time (see Figure 3). The best stability of the prepared emulsions was observed for the system based on interesterified lipids containing a 22–26% polar fraction. The change in consistency of the investigated emulsion under rest conditions, manifested by a variation in G′ modulus at low frequencies, correlates moderately with the content of polar fraction in the structured lipids. In the case of the EM3 emulsion prepared from Blend 3, nearly a 60% increase in G′ modulus was observed after 8 weeks of storage. In turn, for EM2 and EM4 emulsions, the changes were much smaller and did not exceed ~5% in both cases, although in both emulsions, the opposite trend was visible: a decrease in G′ for EM2 and an increase in EM4. For emulsion EM5, based on Blend 5 with the highest amount of polar fraction, the G′ modulus was found to be about 37% smaller in comparison to the values measured for the as-prepared emulsion. The G′ modulus of reference emulsion EM1 increased about 20% after storage. Analysis of the frequency sweeps of G′ alone did not allow for a clear explanation of the observed phenomena, but analysis of the mutual relations of the G′ and G″ moduli values, measured at the same frequency ranges, provided valuable information on the consistency of the investigated materials. Emulsions, from the point of view of linkages formed between interacting droplets, are considered as strain-dependent networks. Characteristic features of such systems are a higher dependence on frequency for the dynamic moduli and smaller differences between the corresponding moduli. In our case, the difference between storage and loss moduli for the as-prepared emulsions were within the range of 30–38%, which broadened after aging to 27–39%.

Emulsions EM2 and EM4 did not show significant changes (less than 5%) in G′ values, while, for EM1—a change from 30% to 27% after aging, and for EM3—a drop from 38% to 29% after 8 weeks of storage in corresponding moduli values were observed. These changes indicate severe modification of the mechanical properties of the material during prolonged storage and the transformation of the microstructure to the strain-dependent networks, which are extremely sensitive to strains and flow upon relatively small stresses, exhibiting more liquid-like behavior. To show the stability of the microstructure, amplitude sweeps within a wide strain amplitude range were investigated and are shown in Figure 4. Rheological tests are simple and fast methods of analysis of aging in emulsion systems and are widely employed in the pharmaceutical and cosmetics industry, allowing comparison of the stability of the formulated emulsions without the need of long-term storage tests [39]. Here, the rheological measurements were performed with at least five repetitions to assure adequate accuracy and reliability of the results. Oscillatory measurements shown in Figure 2 evidence well that all the studied systems exhibit predominantly elastic behavior

after their manufacture and 8 weeks of storage, with values of G' dominating over G". The aging of emulsions was reflected in the changes in the values of storage modulus G' that can be treated as a qualitative (and also quantitative) measure of changes in the emulsion consistency at every stage of production and storage. Few of the investigated systems showed a considerable decrease in their moduli with aging within the whole frequency range, while in others, the increase in G' was noted, pointing to the increased brittleness of the emulsion after storage.

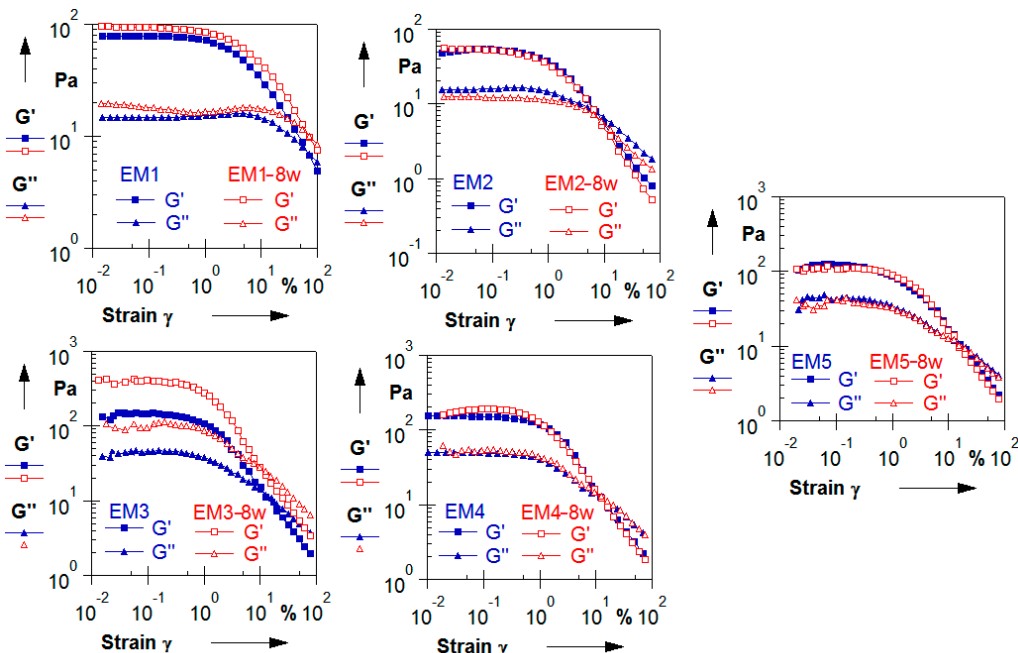

**Figure 4.** G' and G" dependencies on strain amplitude, measured at constant angular frequency 1 rad/s. Error bars for the measured values are within the size of point markers.

Three of the prepared emulsions with interesterified fats (EM2, EM4, EM5) show a stable microstructure within the linear viscoelastic range. The flow curves of the as-prepared and stored materials, recorded at 20 °C in a constant shear stress mode, are shown in Figure 5.

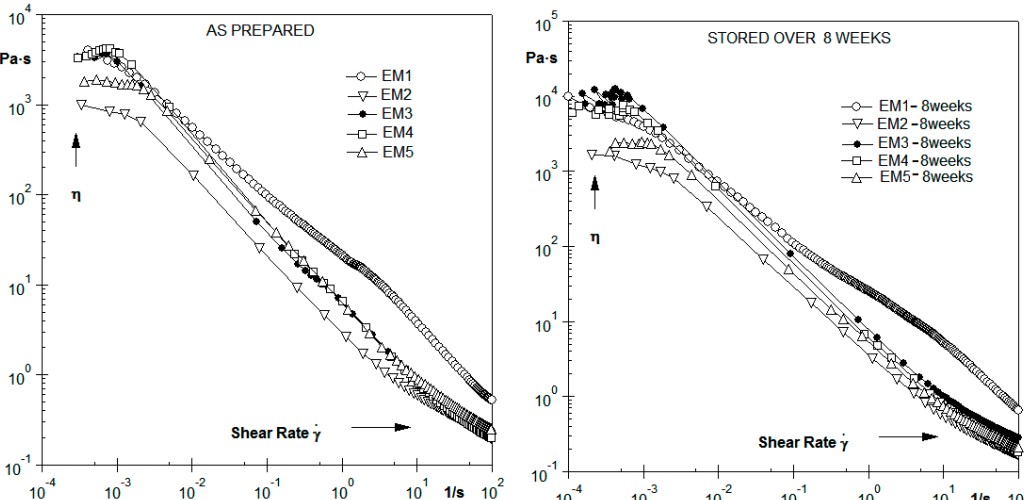

**Figure 5.** Viscosity curves recorded for the as-prepared and stored emulsions with different amounts of polar fraction in interesterified fat (the content of polar fraction has been given in Table 1). Error bars of the measured values are depicted with the markers size.

Qualitatively, the flow behavior of all the emulsions was similar with a distinct plateau region at low shear rates with almost constant zero shear viscosity ($\eta 0$), with the second region characterized by a continuous decrease in viscosity. The appearance of the plateau at low shear rates corresponds to the formation of a stable microstructure under steady conditions due to the enhanced interactions between droplets. The applied mode of measurement enabled the determination of the yield points of all the materials. The corresponding data are collected in Table 2.

**Table 2.** Apparent yield point values and maximum strains of the investigated materials compared to mean droplet size and K indexes.

| Emulsion | Apparent Yield Point | Maximum Strain | Mean Droplet Size | K Index |
|---|---|---|---|---|
| | [Pa] | [%] | [μm] | - |
| EM1 | 5.55 ± 0.39 | 43.8 ± 1.6 | 4.2 a ± 0.01 | 1.8 |
| EM1–8 weeks | 6.31 ± 0.51 | 30.5 ± 2.1 | 6.9 a ± 0.01 | 1.8 |
| EM2 | 1.65 ± 0.23 | 22.7 ± 1.3 | 30.1 d ± 2.25 | 4.4 |
| EM2–8 weeks | 2.40 ± 0.19 | 23.9 ± 1.1 | 39.4 d ± 3.12 | 6.9 |
| EM3 | 3.26 ± 0.31 | 9.6 ± 1.5 | 5.1 c ± 0.01 | 1.8 |
| EM3–8 weeks | 6.98 ± 0.33 | 12.3 ± 0.9 | 15.6 c ± 1.56 | 2.3 |
| EM4 | 4.53 ± 0.26 | 13.9 ± 1.1 | 4.7 b ± 0.02 | 1.8 |
| EM4–8 weeks | 5.52 ± 0.41 | 13.5 ± 1.2 | 14.9 b ± 1.12 | 2.3 |
| EM5 | 4.20 ± 0.35 | 28.6 ± 1.4 | 4.9 bc ± 0.03 | 1.7 |
| EM5–8 weeks | 3.95 ± 0.29 | 19.7 ± 1.6 | 15.5 c ± 0.07 | 2.2 |

a, b, c, d: different letters indicate mean values that differ statistically significantly ($p < 0.05$).

The apparent yield stress of the investigated emulsions usually increased during prolonged storage time, which supports the statements about the microstructure stiffening within longer periods of time. Relatively small changes, both in apparent yield stress and maximum strain, were observed for emulsions with a moderate amount of polar fractions. Maximum strain is a strain value at the flow point and can be understood as an indicator of the emulsion stability at oscillatory deformations. In this sense, we can treat it as a convenient tool for the anticipation of emulsion stability during transportation, where the oscillatory deformations appear and may cause emulsions destabilization. Basically, the emulsions with interesterified fats appear "more rigid" than the conventional emulsion EM1, which is evidenced by lower values of maximum strain. This may indicate a weaker attraction between individual droplets in the novel emulsions due to the insufficient amount of emulsifier in the system.

## 4. Conclusions

It was shown that structured lipids, formed by enzymatic modification of hard fats with selected vegetable oils, are capable of stabilizing emulsion systems without the need to add other synthetic emulsifiers. For E2 emulsions where the amount of mono and diacylglycerols in the system was lower, a poorer emulsion quality was observed. The parameters of average particle size and dispersion coefficient as well as rheological properties indicate that the emulsion requires further development in the field of either adding another emulsifier or changing the viscosity modifier that will synergistically interact with the lower amount of emulsifiers produced during interesterification. We believe that the work on the composition of this type of emulsion, which, at the moment, shows an acceptable emulsifying capacity of prepared esterified fats, will allow us to refine the composition of emulsions, which will bring progress in this field. Generally, emulsions with higher amounts of polar fractions (EM3, EM4, and EM5) are sufficiently stable and can be a proposal for cosmetic bases. Thus, based on the results obtained, it can be believed that the use of new interesterified fats for the preparation of cosmetic emulsions allows

the synthesis of stable systems without synthetic emulsifiers. Rheological measurements performed in shear and oscillatory modes indicate a satisfactory stability of the obtained systems, comparable to lecithin-stabilized emulsions.

**Author Contributions:** Conceptualization, M.K.; methodology, M.K. and A.K.-M.; validation, M.K.-M.; formal analysis, J.S.; investigation, M.K., A.K.-M. and M.K.-M.; data curation, A.Z.; writing—original draft preparation M.K. and A.K.-M., writing—review and editing, A.Z.; visualization, J.S. and A.Z.; supervision, A.Z.; funding acquisition, J.S. All authors have read and agreed to the published version of the manuscript.

**Funding:** This research received no external funding.

**Institutional Review Board Statement:** Not applicable.

**Data Availability Statement:** Not applicable.

**Acknowledgments:** This research was co-financed from the subsidy granted to the Kazimierz Pulaski University of Technology and Humanities in Radom, Poland, Warsaw University of Technology (Warsaw, Poland), Warsaw University of Life Sciences-SGGW, Poland and Cracow University of Economics.

**Conflicts of Interest:** The authors declare no conflict of interest.

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
