# Peer review of "Rheological Characterization and Quality of Emulsions Based on Fats Produced during the Reaction Catalyzed by Immobilized Lipase from Rhizomucor Miehei"

_catalysts, doi:10.3390/catal12060649_

Round 1

Reviewer 1 Report

I think the author has made positive changes as suggestions, which are of great significance to improve the quality of the article. So I think it can be accepted.

Author Response

Dear Reviewer,

 Thank you for the accepation of our work.

Regards,

Reviewer 2 Report

The manuscript of Kowalska and co-workers describes a set of emulsions containing interesterified fat blends oily phase with enhanced amounts monoacylglycerols (MAGs) and diacylglycerols (DAGs), obtained through enzymatic interesterification (as one expects from such a kind of transformation). Substantially this work lacks of sufficient innovative contents, the authors reports with useless amount of work a kind of application which is routinely conduced even at the Food and Cosmetic raw materials companies (i.e. the French Company Gattefossè).

The whole manuscript is confusing, the huge amount of data distracts the reader from the real focus. There are several fats already in the marked obtained by the same technology and performing as well as that described. One could write two papers from this work directed to Editors of industrial Journals (i.e. H&PC Today or Personal Care Magazine).

Efficiency of the obtained emulsifiers is not so proven indeed.

Basically, the work is a repetition of a previous work conducted by the same authors in International Journal of Cosmetic Science (2014) https://doi.org/10.1111/ics.12173; Rheologica Acta (2020) doi: 0.1007/s00397-020-01232-6. Or similar applications (see Acta Pol Pharm 2017).

Author Response

Dear Reviewer,

Thank you for referring to our article and thus to our work. However, our project is very broad.  It is about testing the behaviour of different modified fats as a fat emulsion base and then testing the properties of new emulsion products. We are focusing on different properties including a full commodity evaluation, which is very important and useful from the consumer's point of view.

We evaluate functional, physicochemical and sensory properties and conduct consumer research. The reviewer pointed out three previous papers in which he thought our reports had been published before. Unfortunately in my opinion these are quite different papers. In the International J of Cosmetic Science the focus was on sensory evaluation studies and skin moisturizing properties. In a second journal, R Acta, the results of the study concerned a completely different fat, primarily rabbit and pumpkin seed oil. In the next article cited by the reviewer, the work also concerned sensory evaluation. Each of our work is a continuum of some stage, unfortunately we do not manage to publish it all chronologically.

The work presented and submitted to Catalysts deals primarily with the behaviour of emulsions and, more specifically, the determination of their rheological properties. Obtaining a fat with the right amount of emulsifiers is not an easy task at all. Fat is a natural product, so each repetition must be carried out on new samples. Interestrification of different fats produces a new fat which is unknown in nature. We have noticed that the emulsifiers we have obtained have different efficiencies, so we are working on confirming this (we have not published this yet, so I am only signalling). It is related to the type of modified fat, i.e. its final structure and distribution of the appropriate fatty acids in the SN positions. And exactly it concerns the structure of mono and diacylglycerols. But this does not mean, as the reviewer wrote, that the effectiveness is unproven. Besides, our other works also show synergism of these emulsifiers with properly selected rheology modifiers.

I consider the statement by the reviewer of our work as useless material unjustified. Perhaps in the company indicated by the reviewer such fats are produced, but whether such fats containing in its composition also an emulsifier???? Interesrification is actually used to modify fats and is increasingly used, if only because of the food regulations that came into force in 2021 which stipulated the complete removal of trans fats from food products.  This process guarantees products free of trans isomers and has therefore become an alternative to partial hydrogenation.

As for dividing the material into two articles, in my opinion I don't really know how or what should be included in each article. What data? I think this is a very individual approach to this issue...

Regards,

Reviewer 3 Report

This report investigates the use of modified fats in the preparation and stabilization of emulsions

 The experimentation was done in a systematic manner and the report was written fairly well. The manuscript can be further improved after minor revisions suggested below:

  • The title refers to a qualitative analysis of the emulsions. However, a quantitative analysis was performed because numerical variables were analyzed and a relationship was established between these analyses. I suggest suitability of the title.
  • In the methods section, line 191 page 4 the author mentions that after cooling the emulsions were stored in the refrigerator. At what temperature? For how long? Why was the stability of emulsions investigated only by storing these emulsions at low temperatures?
  • I suggest statistical analysis of all data as well as expressing the results as mean and standard deviation. In the discussion, emphasize whether the differences between the variables studied are statistically significant or not.
  • On line 285 page 7 explain how the storage condition may have interfered with the globule size of the emulsions.
  • On line 299 page 7 the paragraph is the same as on line 273 page 6.
  • Review article writing. In line 472, page 12 you can see the expression " de facto".

Author Response

Dear Reviewer,

Thank you very much for your guidance to improve the content and quality of our work. Below I have referred to the suggestions and remarks indicated by the reviewer. The changes to the work were made in the change tracking file.

Suggestion 1;

The experimentation was done in a systematic manner and the report was written fairly well. The manuscript can be further improved after minor revisions suggested below: 

The title refers to a qualitative analysis of the emulsions. However, a quantitative analysis was performed because numerical variables were analyzed and a relationship was established between these analyses. I suggest suitability of the title.

Answer 1

The title was amended as follows: Rheological characterization and quality of emulsions based on fats produced during the reaction catalysed by immobilised lipase from Rhizomucor miehei

Suggestion 2;

In the methods section, line 191 page 4 the author mentions that after cooling the emulsions were stored in the refrigerator. At what temperature? For how long? Why was the stability of emulsions investigated only by storing these emulsions at low temperatures?

Answer 2;

The emulsions were stored at 8-10C. From previous research and experience, the authors concluded that these were the optimum conditions for storing this type of product, hence in this paper they are used.

Suggestion 3;

I suggest statistical analysis of all data as well as expressing the results as mean and standard deviation. In the discussion, emphasize whether the differences between the variables studied are statistically significant or not.

Answer 3

The statistical analysis has been introduced

Suggestion 4;

On line 285 page 7 explain how the storage condition may have interfered with the globule size of the emulsions.

Answer 4

The paragraph has been added

Suggestion 5

On line 299 page 7 the paragraph is the same as on line 273 page 6.

Answer 5

One paragraph has been removed.

Suggestion 6

Review article writing. In line 472, page 12 you can see the expression " de facto".

Answer 6

The expression " de facto" has been removed.

Regards,

This manuscript is a resubmission of an earlier submission. The following is a list of the peer review reports and author responses from that submission.

Round 1

Reviewer 1 Report

The authors showed the rheological properties of emulsions including interesterified fats for food and cosmetic applications. However, there is no innovative information in this manuscript. Therefore, this manuscript is not suitable for publication. The reviewer shows some comments for this manuscript.

  1. The authors described "however the parameters of the final products are not yet optimized as it can be seen from the obtained results." in Conclusions. The reviewer also thinks that. There is no innovative information.
  2. Entirely, there is no detailed discussion for individual samples. The authors should describe why such data was obtained, corresponding to the results for each sample. The authors only showed concise and abstract discussion.
  3. If plural graphs are shown in one Figure, we should label each graph, for example, (a), (b), (c), ...... . In all figures, the graphs are not labelled.
  4. In figure 1, there is a lot of information; therefore, the readers may confuse. If the author would like to compare the samples EM1-5, the authors should show the same kind of data corresponding to EM1-5 in one graph.

Reviewer 2 Report

The manuscript of Kowalska and co-workers deals with rheological studies of emulsions based on emulsifiers produced during the interesterification process and containing Hydroxypropylmethylcellulose.

No doubt on the amounts of data presented in this study.

However, I will rapidly go to the point.

The authors in continuity with a previously published paper from the same group (“Rheological and physical analysis of oil-water emulsion based on enzymatic structured fat”.( Rheologica Acta (2020) 59:717–726; doi: 0.1007/s00397-020-01232-6) propose the identical main message with an identical approach, just by exchange the type of oil.

Additionally, the authors claim in the conclusions section that; “The interesterification process leads to the formation of new emulsifiers, which are able to form relatively stable emulsions, however the parameters of the final products are not yet optimized as it can be seen from the obtained results”. 

The messages reported in the abstract section appears vague and redundant.

As a whole I regret, but in my opinion, this article does not give the necessary conceptual novelty to the readers and thus is not suitable for publication.

Reviewer 3 Report

This study is aiming to produce the structured lipids, generated in enzymatic interesterification of natural hard fats with selected vegetable oils, and its effect on stabilizing emulsion systems. However, in general, this study is relatively simple and the experimental data are not complete enough. Besides that, the discussion and analysis of the experimental phenomena are not deeply.

  • In the article, the interesterification process leads to the formation of new emulsifiers. I think authors should provide more data and show the reaction equation.
  • No data can prove that the emulsion has been prepared. The authors should provide the microstructure picture of emulsion or others which can explain the microstructure of emulsion.
  • Why the rheology can be used to analyze the stability of emuslion? I think it have to figure out the relationship between the rheological properties and the microstructural changes of emulsion, produced in this article, with aging, firstly. Otherwise, I think this method is not very rigorous.